# Enhancing phosphate-solubilising microbial communities through artificial selection

Lena Faller [1,2], Marcio F. A. Leite [1,2] & Eiko E. Kuramae [1,2]

Microbial communities, acting as key drivers of ecosystem processes, harbour immense potential for sustainable agriculture practices. Phosphate-solubilising microorganisms, for example, can partially replace conventional phosphate fertilisers, which rely on finite resources. However, understanding the mechanisms and engineering efficient communities poses a significant challenge. In this study, we employ two artificial selection methods, environmental perturbation, and propagation, to construct phosphate-solubilising microbial communities. To assess trait transferability, we investigate the community performance in different media and a hydroponic system with *Chrysanthemum indicum*. Our findings reveal a distinct subset of phosphate-solubilising bacteria primarily dominated by *Klebsiella* and Enterobacterales. The propagated communities consistently demonstrate elevated levels of phosphate solubilisation, surpassing the starting soil community by 24.2% in activity. The increased activity of propagated communities remains consistent upon introduction into the hydroponic system. This study shows the efficacy of community-level artificial selection, particularly through propagation, as a tool for successfully modifying microbial communities to enhance phosphate solubilisation.

In agriculture, the addition of plant growth-promoting microorganisms offers a sustainable alternative to conventional crop production by reducing the need for agrochemicals and pollutive fertilisers derived from finite resources[1,2]. Microbial communities play a vital role in various ecosystem processes and have diverse applications in agriculture, including protection against environmental stresses, increased pathogen resistance, and enhanced nutrient acquisition[3–5]. Phosphate-solubilising microorganisms, for instance, improve the accessibility and uptake of phosphorus by releasing plant-available phosphate from fixed sources[6]. To harness the metabolic diversity of these beneficial microorganisms, the use of microbial consortia has gained interest over single isolates[7].

However, constructing, optimising, and applying beneficial microbial consortia pose significant challenges[8–10]. The transfer of these consortia from laboratory to field settings is hindered by the resistance and resilience of the target microbiome to inoculants and the decreased interaction and co-dependence between plants and beneficial microorganisms due to high nutrient inputs[8,10,11]. Additionally, the intricate interactions among community members, which are difficult to predict, can lead to incompatibility and unstable functioning[8]. Consequently, the conventional bottom-up approach to constructing microbial consortia by combining the best candidates may result in decreased functionality, rendering it inefficient. An alternative approach for consortium construction is the artificial selection of microbial communities[12]. This top-down approach starts with the entire community, ensuring compatibility by linking changes in composition and functions[13]. Moreover, it allows for the measurement of total community function, capturing the metabolic diversity of plant growth-promoting traits[12]. For instance, microorganisms employ various phosphate solubilisation strategies, such as exuding organic or inorganic acids[14,15], releasing $H^+$ ions into the environment[16], or producing siderophores[17]. Artificial selection of microbial communities has been conducted either in vitro or in presence of the host microbiome[12,13]. Host-mediated artificial selection has shown promise

[1]Department of Microbial Ecology, Netherlands Institute of Ecology (NIOO-KNAW), Droevendaalsesteeg 10, 6708 PB Wageningen, The Netherlands. [2]Utrecht University, Institute of Environmental Biology, Ecology and Biodiversity, Padualaan 8, 3584 CH Utrecht, The Netherlands. ✉e-mail: e.kuramae@nioo.knaw.nl

in engineering plant microbiomes for beneficial traits like salt tolerance[18] or drought tolerance[19], altered flowering time[20,21], and increased plant biomass[22]. While host-mediated artificial selection holds great potential, in vitro artificial selection offers a better starting point for the large-scale production of microbial inoculants[23]. To date, in vitro selection has been applied to shift $CO_2$ emission[24], enhance amylolytic activity[25], or reduce toxin levels[26], but its application to optimise plant growth-promoting consortia remains unexplored. Various methods of in vitro artificial selection have been proposed, including propagation, mixed propagation, environmental perturbations, and the addition or knockout of specific species[13,22,26]. However, our understanding of the efficiency of these methods in optimising community functioning remains limited.

In this study, we aim to compare two in vitro artificial selection methods - environmental perturbation and propagation - to optimise the plant growth-promoting trait of phosphate solubilisation. Environmental perturbation exposes microbial communities to external selection pressures, mimicking gradual natural evolution[12,13]. In contrast, propagation involves the repeated seeding of new offspring communities from random subsets of each selected community based on the desired community function, favouring high-performing communities through intercommunity selection[27,28]. Moreover, this study aims to investigate the ecology of a cultivable phosphate-solubilising community by combining dilutions and soil community fractioning. Fractioning by repeated Nycodenz extraction has been proposed to result in communities with varying degrees of association with the soil matrix, while serial dilution provides insights into how the complexity of the community affects phosphate solubilisation[29,30]. We address the following questions: (i) Which fraction of phosphate solubilisers within a soil microbial community is ecologically relevant? (ii) Which in vitro selection method is more efficient in increasing microbial phosphate solubilisation activity? (iii) Can the performance of the selected communities be maintained and transferred to different environments? To achieve these objectives, our study employs an interdisciplinary approach that integrates cultivation and selection of phosphate-solubilising communities with 16S rRNA gene amplicon sequencing, and post-selection experiments, using a hydroponic system with a commercial chrysanthemum (*Chrysanthemum indicum* L.) cultivar. The cultivation of chrysanthemums heavily relies on inorganic nutrient input, prompting the need to explore alternative fertilisation strategies[31]. Our findings in this study demonstrate that the propagation method yields microbial communities with a significantly greater increase in phosphate solubilisation compared to environmental perturbation, and this enhanced solubilisation capacity is transferable across diverse environmental contexts.

## Results

### Ecology of the source soil microbial community

To explore different soil matrix association strengths, we performed repeated Nycodenz extraction to create three fractions, easy-to-extract (EE), hard-to-extract (HE), and not-extractable (NE), that allowed us to explore the ecologically relevant fraction of phosphate solubilisers within a soil microbial community (Fig. 1i)[30]. The fractioned communities did not differ significantly in their amount of solubilised phosphate (one-way Analysis of Variance (ANOVA), $F(2,6) = 0.217$, $p = 0.811$; Supplementary Table 1; Fig. 2a). NE communities exhibited a lower variability in the amount of solubilised phosphate spanning a range of $3.0\,\mu g\,P\,ml^{-1}$, compared to EE communities (range $= 60.5\,\mu g\,P\,ml^{-1}$) and HE communities (range $= 39.3\,\mu g\,P\,ml^{-1}$), indicating a more consistent phosphate solubilisation activity within the NE communities. Analysis of the 16S rRNA amplicon sequencing revealed that the EE and HE communities were similar, whereas the NE communities differed (Fig. 2b). The EE and HE communities showed significant positive shifts in *Novosphingobium* and *Pseudomonas*. In contrast, the NE communities had a significant positive shift in *Klebsiella*, and no *Novosphingobium* were detected.

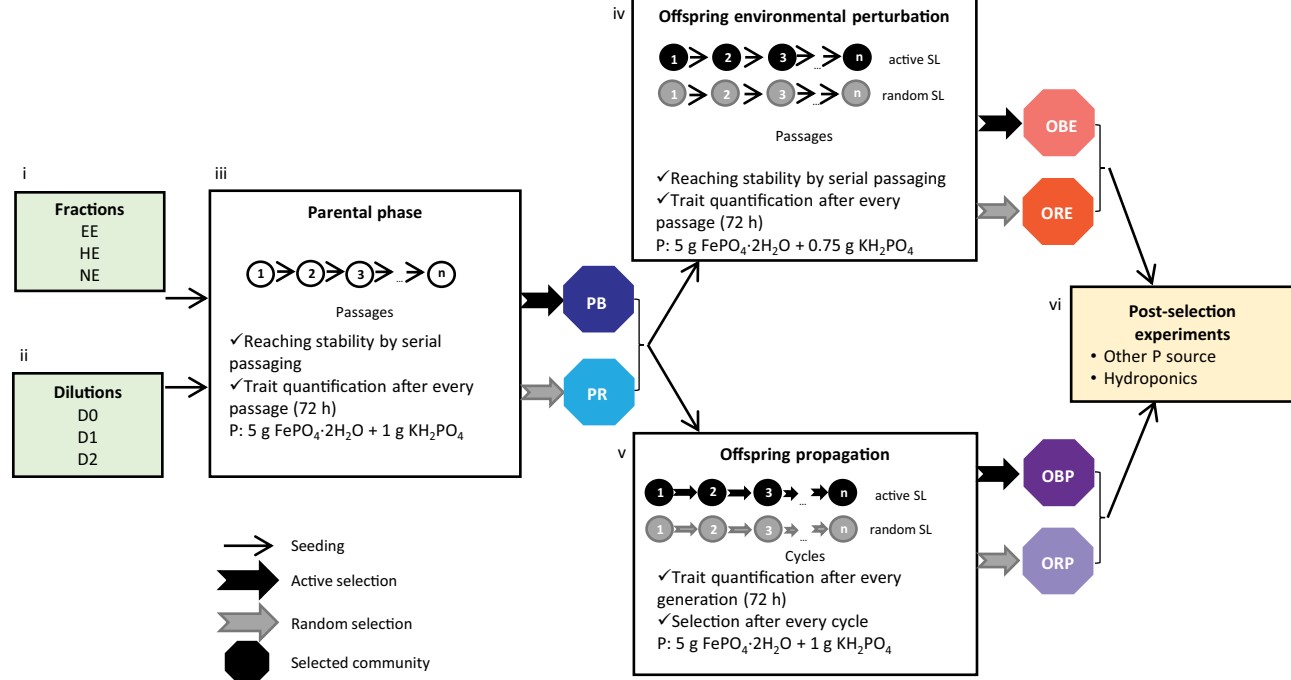

**Fig. 1 | Experimental workflow.** (i) Communities of the Nycodenz extraction. Fractions: EE = easy-to-extract, HE = hard-to-extract, NE = not-extractable. (ii) Dilution of communities: D0 = undiluted, D1 = 1:2 (v:v), D2 = 1:4 (v:v). (iii) Communities of (i) and (ii) were used to start the parental phase, trait stabilisation by repeated passaging (72 h = 1 passage), and selection of the best parental (PB) and random parental (PR) communities. The selected communities seeded active and random selection lines (SL) in (iv) environmental perturbation or (v) propagation selection experiments. The best offspring (OBE & OBP) and random offspring (ORE & ORP) communities of the methods (iv) and (v) were selected. (vi) The post-selection experiments compared the performance of the selected communities.

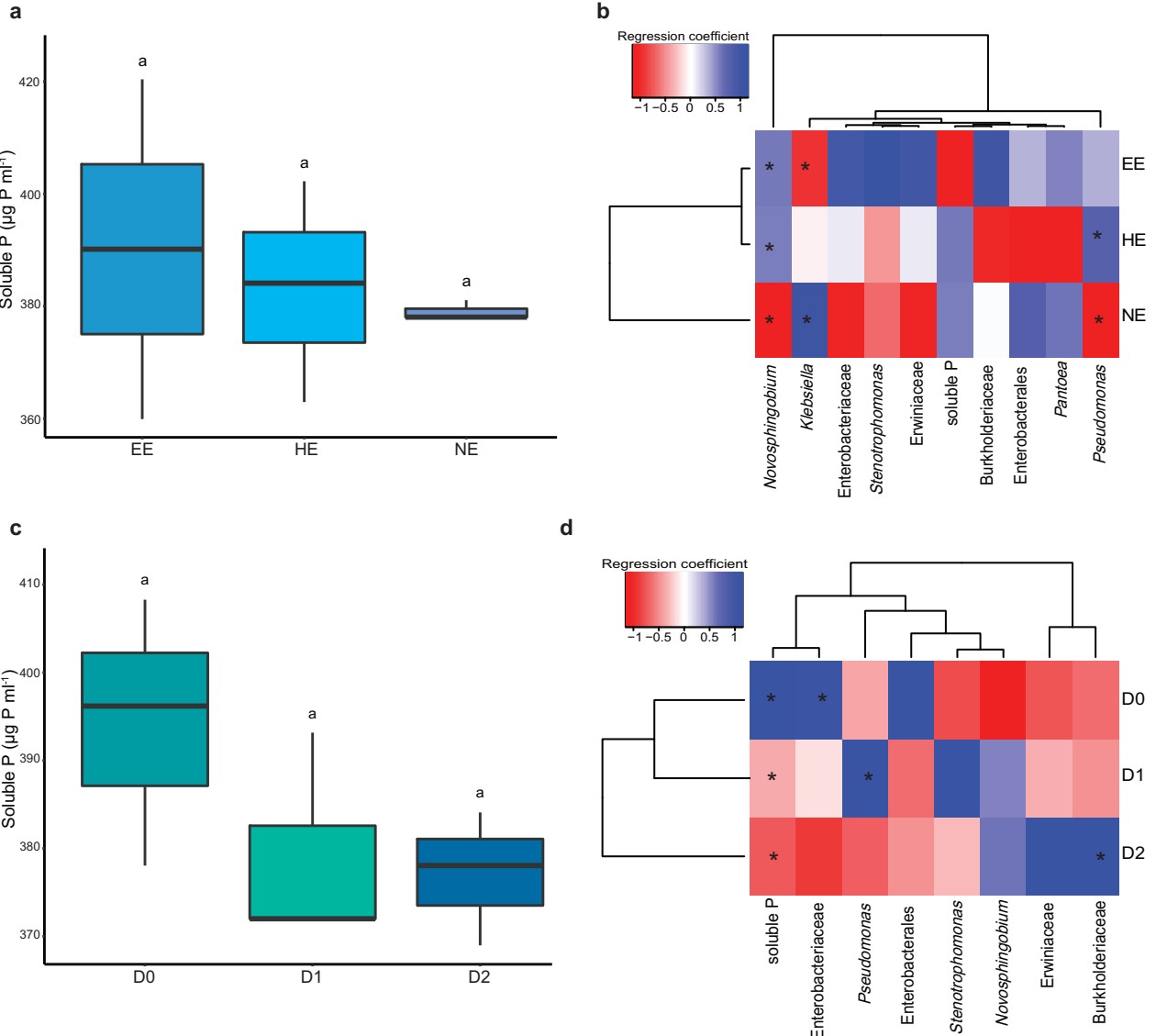

**Fig. 2 | Ecology of the source soil microbial community. a** Solubilised phosphate by the fractioned communities, **b** The regression coefficients of the fractioned communities, **c** Solubilised phosphate by the diluted communities, **d** The regression coefficients of the diluted communities. Communities: EE: easy-to-extract ($n = 3$ communities), HE: hard-to-extract ($n = 3$ communities), NE: not-extractable ($n = 3$ communities), D0: undiluted ($n = 3$ communities), D1: 1:2 (v:v, $n = 3$ communities), D2: 1:4 (v:v, $n = 3$ communities). The shifts in regression coefficients are blue for positive, white for non-significant, and red for negative shifts. Community members are represented as ASVs and labelled with the most specific known taxonomic level. Asterisks indicate significant differences between communities.

Significance for the regression coefficient from GJAM obtained by the credible interval after the 10,000 MCMC (Markov Chain Monte Carlo) iterations to estimate the values for each regression coefficient; whenever zero is within this 95% credible interval the regression coefficient is not considered as significant. Box plots: box spans 25th to 75th percentiles (Interquartile Range, IQR) with median as central line, whiskers cover values within 1.5 times IQR. Letters in box plot: groups with different letters are statistically different based on pairwise comparison ($p$ values adjusted via Benjamini-Hochberg procedure). Source data are provided as Source Data file.

We further diversified the communities by combining EE and HE communities to create an undiluted community (D0) that was then diluted by 1:2 (D1) and 1:4 (D2) to establish a link between community complexity and phosphate solubilisation (Fig. 1ii). The diluted communities did not show a significant difference in their activity and variability in phosphate solubilisation (one-way ANOVA, $F(2,6) = 1.799$, $p = 0.244$; Supplementary Table 2; Fig. 2c). However, the analysis performed with GJAM showed that the D0 communities exhibited a significant increase in the concentration of soluble phosphate (with an average of $394.2 \pm 15.2 \, \mu g \, P \, ml^{-1}$; Fig. 2d), which changed to a negative coefficient for D1 and D2 indicating that the phosphate solubilisation decreased following dilutions. Diluting also significantly shifted the phosphate-solubilising communities towards one dominant amplicon

sequence variant (ASV) (Fig. 2d). The D0 communities were dominated by an ASV from Enterobacteriaceae family, the D1 communities by a *Pseudomonas* sp., and D2 communities by an ASV from Burkholderiaceae family.

### Parental phase

In the parental phase (Fig. 1iii), the solubilisation activity of the communities did not significantly differ among the different fractioned and diluted source communities (one-way ANOVA, $F(5,184) = 1.478$, $p = 0.199$; Supplementary Table 3; Fig. 3a, b), but it oscillated across the passages. Among the fractioned communities, EE had the most phosphate-solubilising communities (38.5%) that significantly increased the amount of soluble phosphate compared to the sterile

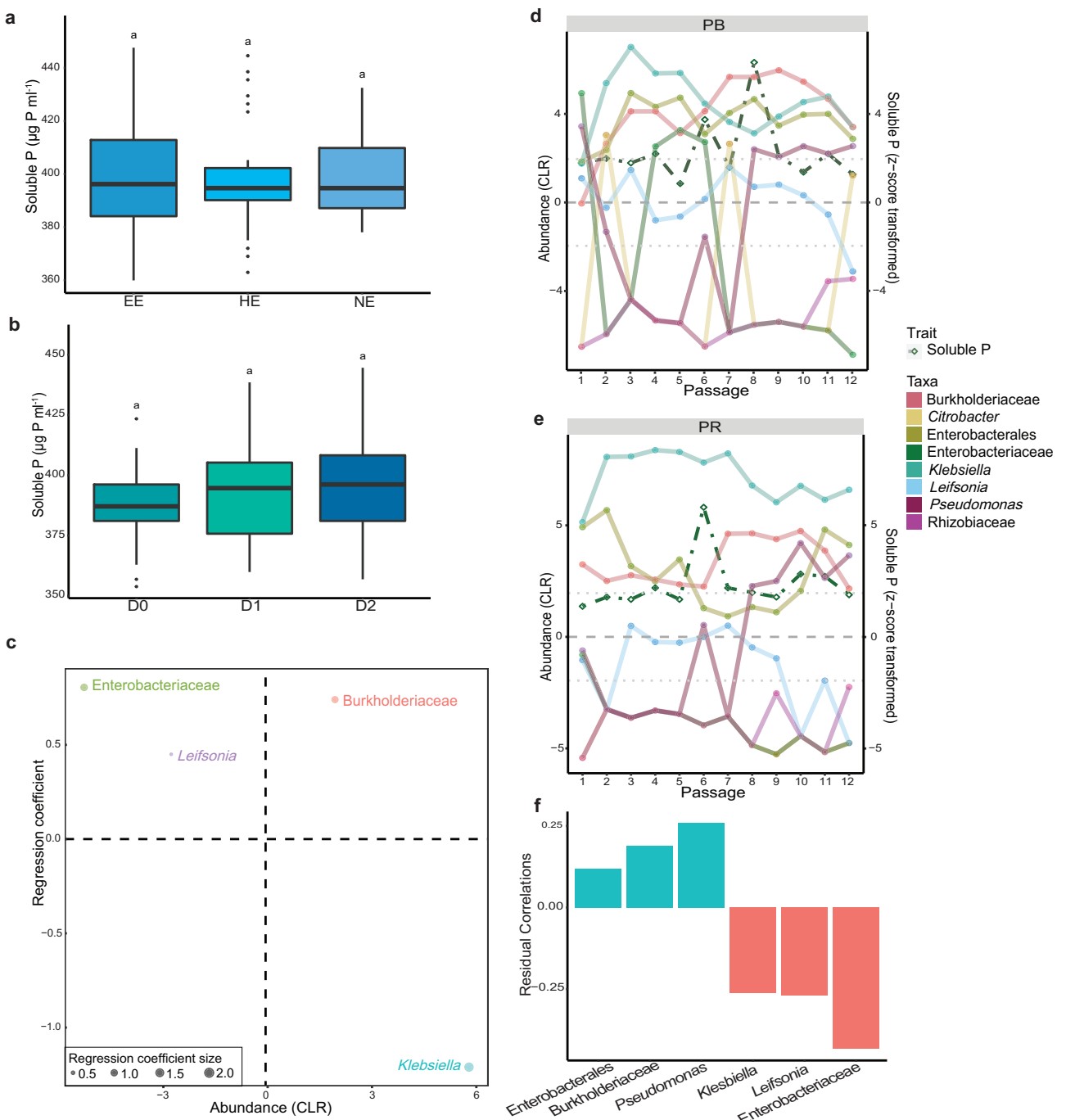

**Fig. 3 | Parental phase. a** Solubilised phosphate across the total parental phase by the fractioned communities. **b** Solubilised phosphate across the total parental phase by the diluted communities. **c** Regression coefficients comparing the best-performing and the randomly selected parental communities. Abundances are centre log-ratio (CLR) transformed. **d**, **e** Community composition of the selected communities throughout the parental phase per passage showing the abundance (CLR). **f** Residual correlations of the community members linked to solubilised phosphate. Community members represented by ASVs and labelled with the most specific known taxonomic level. Box plots: box spans 25th to 75th percentiles (IQR) with median as central line, whiskers cover values within 1.5 times IQR, and outliers as dots beyond whiskers. Letters in box plot: groups with different letters are statistically different based on pairwise comparison (*p* values adjusted via Benjamini-Hochberg procedure). Fraction communities: EE: easy-to-extract (*n* = 36 communities), HE: hard-to-extract (*n* = 36 communities), NE: not-extractable (*n* = 36 communities). Diluted communities: D0: undiluted (*n* = 36 communities), D1: diluted 1:2 (v:v; *n* = 36 communities), D2: diluted 1:4 (v:v; *n* = 36 communities). Source data are provided as Source Data file.

control (z-score > 1.96). The NE communities had the highest average soluble phosphate concentration (407.4 ± 29.9 µg P ml⁻¹). In terms of dilutions, D2 communities displayed the highest average phosphate solubilisation (404.8 ± 33.9 µg P ml⁻¹), with 43.6% of D2 communities showing a significant increase of soluble phosphate compared to the sterile control (z-score > 1.96). In contrast to the source community,

D0 communities exhibited the lowest phosphate solubilisation activity.

After approximately eight passages, the oscillation in soluble phosphate decreased, indicating stabilisation. Within this island of stability, the community of EE passage 8 was the best-performing community (PB) with an increase in the soluble phosphate

concentration by 285% (523.3 µg P ml⁻¹) compared to the first passage. The randomly selected community (PR), NE passage 10, solubilised 64.6% (420.4 µg P ml⁻¹) more phosphate compared to the first passage. The selected communities also differed in their community composition. The best-performing community had the highest diversity (Shannon index = 1.05 ± 0.12) and the greatest abundance of *Leifsonia*, Burkholderiaceae and Enterobacteriaceae (Fig. 3c). On the other hand, the randomly selected communities had the lowest diversity (Shannon index = 0.21 ± 0.08) and was highest in *Klebsiella*.

The abundance of the community members revealed distinct patterns in the phosphate-solubilising microbial communities. Within the PB community, ASVs of Burkholderiaceae family, *Klebsiella* genus, and Enterobacterales class were consistently abundant across passages (Fig. 3d). ASVs belonging to *Pseudomonas*, *Citrobacter*, *Leifsonia*, and Enterobacteriaceae displayed pronounced oscillations. In the PR community, *Klebsiella* ASV was consistently the most abundant throughout the passages, followed by ASVs of Burkholderiaceae and Enterobacterales (Fig. 3e). Furthermore, residual correlations analysis showed positive associations between soluble phosphate concentrations and ASVs of Enterobacterales, Burkholderiaceae, and *Pseudomonas*, while negative correlations were observed with ASVs of *Klebsiella*, Enterobacteriaceae, and *Leifsonia* (Fig. 3f).

## Artificial selection by environmental perturbation

In the environmentally perturbed offspring phase, the selected parental communities were subjected to an environment with reduced phosphate availability, creating an environmental selection pressure (Fig. 1iv). Environmentally perturbed actively selected communities had a similar phosphate solubilisation activity compared to the randomly selected communities (one-way ANOVA, $F_{(1,201)} = 0.057$, $p = 0.812$; Supplementary Table 4; Fig. 4a). Among the active selection line communities, 24.3% outperformed their seeding communities, while surprisingly, 30.6% of the random selection line communities had higher performance than their parental communities. Overall, phosphate solubilisation was significantly affected by the phase (two-way ANOVA, $F_{(1,207)} = 8.093$, $p = 0.005$; Supplementary Table 5). Environmentally perturbed communities showed a significant reduction in phosphate solubilisation (z-score average ± standard error (SEM), 0.8 ± 0.1) compared to the communities of the parental phase (z-score average ± SEM, 1.8 ± 0.5). This difference was confirmed through various statistical analyses, including a two-way ANOVA with Tukey post hoc test ($p = 0.028$), a generalised linear model (GLM) with estimated marginal means (EMMs) ($t(207) = 2.770$, $p = 0.006$), and a linear mixed-effect model (LMM) with EMMs ($t(167) = 3.090$; $p = 0.002$) (Supplementary Table 6; Fig. 4b). The observed reduction corresponds to a 55.6% decrease in phosphate solubilisation in the environmentally perturbed communities compared to the communities of the parental phase. Notably, there was no significant effect of the active or random selection line (two-way ANOVA, $F_{(1,207)} = 0.054$, $p = 0.817$; Supplementary Table 5). Both actively selected communities (z-score average ± SEM, 0.9 ± 0.1) and randomly selected communities (z-score average ± SEM, 0.9 ± 0.1) displayed similar phosphate solubilisation activity relative to the sterile control. Following stabilisation, we selected the offspring best-performing community (OBE), which solubilised 399.2 µg P ml⁻¹ (z-score = 5.3). The offspring randomly selected community (ORE) solubilised 290.3 µg P ml⁻¹ (z-score = 0.3).

Amplicon sequencing analysis revealed that environmental perturbation had a greater impact on the communities compared to the performance of the parental communities (Fig. 4c). Both environmentally perturbed offspring communities showed an increase in *Pseudomonas* and a decrease in Burkholderiaceae abundance, indicating a shared response to the selection pressure. Environmental perturbation shifted the communities towards ASVs of the family Enterobacteriaceae, order *Enterobacteriales*, and the genus *Citrobacter* in the best-performing communities, similar to the PB and PR

communities. ORE communities were enriched in *Klebsiella*, while the best-performing communities exhibited higher diversity (Shannon index = 0.9 ± 0.26) compared to randomly selected ones (Shannon index = 0.31 ± 0.26). Comparison of community member abundances throughout the environmental perturbation phase showed similar patterns between OBE and ORE communities. OBE communities were predominantly abundant in ASVs belonging to *Klebsiella* and Enterobacterales (Fig. 4d), while ORE communities were primarily dominated by *Klebsiella* (Fig. 4e). The communities of the environmental perturbation phase exhibited less oscillation compared to the parental phase communities, suggesting greater stability in community compositions (Figs. 3d, e, 4d, e). In the environmental perturbation phase, network analysis of residual correlations uncovered two competing groups (Fig. 4f). The first group comprised ASVs, including *Klebsiella* and *Pseudomonas*, which exhibited positive correlations with solubilised phosphate. In contrast, the second group, consisting of Enterobacterales, Burkholderiaceae, and Rhizobiaceae, showed negative associations with solubilised phosphate.

## Artificial selection by propagation

In the offspring propagation phase, the communities seeded from the selected parental communities were repeatedly selected to maximise intercommunity selection (Fig. 1v). The propagated communities of the two active selection lines (OBP1 and OBP2) consistently demonstrated significant phosphorus solubilisation compared to the sterile control over cycles (z-score > 1.96) and performed improvements of up to 22.9 % compared to the source community (Fig. 5a). The community OBP2R5 of cycle 6 of the OBP2 selection line was selected as best-performing community of the offspring propagation phase (OBP) with a soluble phosphate concentration of 435.5 µg P ml⁻¹ (z-score = 3.4). The randomly selected community (ORP), cycle 6 community R12 in the randomly propagated selection line, solubilised 332.7 µg P ml⁻¹ (z-score = −1.0). Across the cycles, the OBP communities significantly outperformed ORP communities, as confirmed by multiple statistical analyses: a two-way ANOVA with Tukey post hoc test ($p = 0.014$), a GLM with EMMs ($t(26) = 4.016$, $p = 0.003$), and a LMM with EMMs ($t(14.0) = 3.404$, $p = 0.026$) (Supplementary Table 7; Fig. 5b). Particularly, OBP communities did not significantly differ from the parental communities (Supplementary Table 8). However, they solubilised, on average, 25.0% more phosphate relative to the sterile control compared to the best-performing parental communities. Additionally, further analysis using GLM and LMM with post hoc EMMs revealed that actively selected communities solubilised more phosphate than randomly selected ones. This was evident in the GLM + EMMs ($t(26) = 2.950$, $p = 0.001$) and LMM + EMMs ($t(14) = 2.501$, $p = 0.025$) analyses, although not in the two-way ANOVA with Tukey post hoc test ($p = 0.378$) (Supplementary Table 8).

Propagation resulted in a significant positive shift in the offspring communities towards ASVs of the genus *Citrobacter*, family Enterobacteriaceae, and order Enterobacterales, while reducing the abundance of *Leifsonia* and Burkholderiaceae (Fig. 5c). The OBP communities clustered with the parental communities as subclusters, indicating a combined effect of performance and the selection method on community shifts (Fig. 5d–f). Among the active selection lines, OBP1 communities exhibited the highest diversity (Shannon index = 0.65 ± 0.23), followed by OBP2 communities (Shannon index = 0.48 ± 0.51). In contrast, the ORP communities were less diverse (Shannon index = 0.01 ± 0.4) and dominated by *Pseudomonas* and Burkholderiaceae. The abundances of community members remained relatively stable until the fifth cycle for all selection lines, after which *Pseudomonas* rapidly increased in abundance, leading to changes in community compositions. The actively selected communities were characterised by high abundance of Enterobacterales, while the randomly selected communities were dominated by Burkholderiaceae (Fig. 5d, e). In contrast to the parental phase and offspring

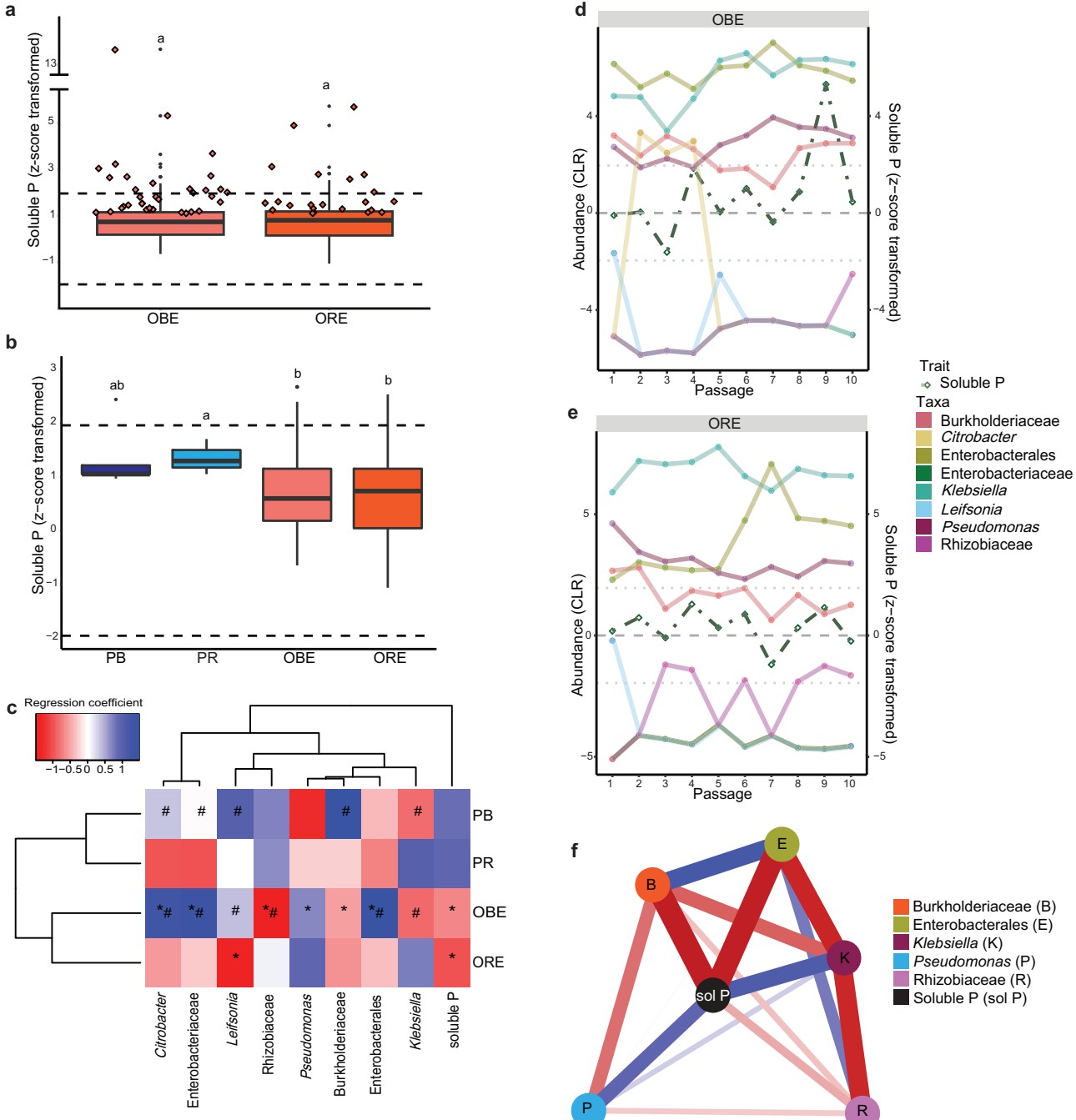

**Fig. 4 | Artificial selection by environmental perturbation. a** Phosphate solubilisation by the stable communities of the best-performing ($n = 144$ communities) and randomly selected ($n = 72$ communities) environmental perturbation offspring phase. Diamonds: communities that solubilised more phosphate than their parental communities. **b** Comparison of the phosphate solubilised by the selected parental communities (PB: $n = 6$ communities, PR: $n = 3$ communities) and the stable environmentally perturbed offspring phase (OBE: $n = 144$ communities, ORE: $n = 72$ communities). **c** Shifts in regression coefficients in communities of the parental and environmental perturbed offspring phase. *: Significances between selection lines. #: Significances between phases. Significance for the regression coefficient from GJAM obtained by the credible interval after the 10 000 MCMC iterations to estimate the values for each regression coefficient; whenever zero is within this 95% credible interval the regression coefficient is not considered as significant. **d**, **e** Community composition and abundance (CLR) of the community members per passage of the selected communities per passage. **f** The residual correlation linked to solubilised phosphate. Relative closeness centrality is the highest for solubilised phosphate. Positive residual correlations shown as blue lines and negative residual correlations as red lines with the line thickness relating to the size of the residual correlations. Box plots: box spans 25th to 75th percentiles (IQR) with median as central line, whiskers cover values within 1.5 times IQR, and outliers as dots beyond whiskers. Letters in box plot: groups with different letters are statistically different based on pairwise comparison ($p$ values adjusted via Benjamini-Hochberg procedure). Communities: PB parental best-performing communities, PR parental randomly selected communities. OBE environmentally perturbed best-performing communities. ORE environmentally perturbed randomly selected communities. Community members represented by ASVs and labelled with the most specific known taxonomic level. Source data are provided as Source Data file.

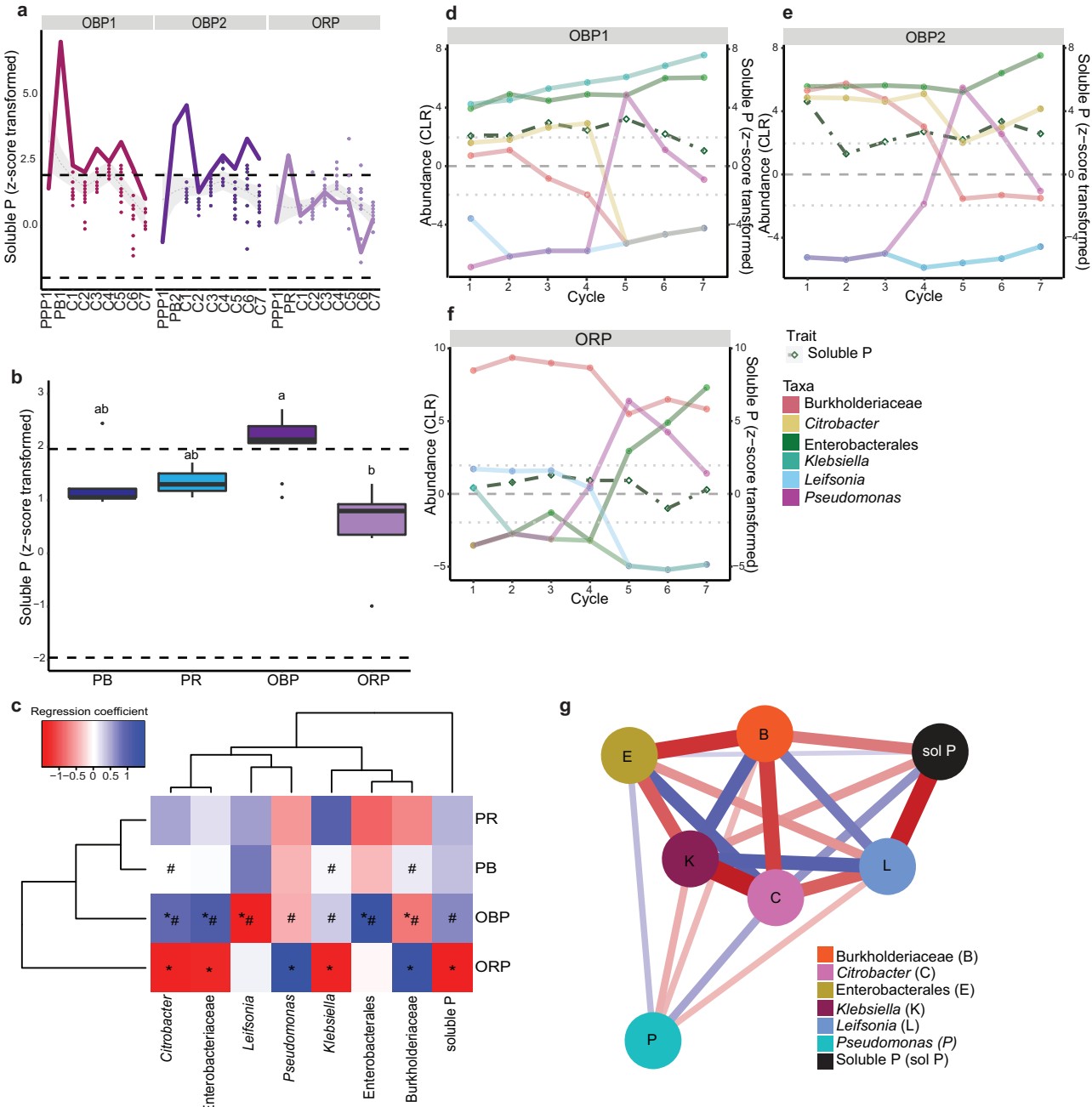

**Fig. 5 | Artificial selection by propagation. a** Phosphate solubilisation by the propagated offspring communities (per selection line $n = 84$ communities). The solid line follows the selected communities. The grey lines show the best fit suggested by the LOESS (locally weighted smoothing approach) with the grey error band as 95% confidence interval. PPP1: Parental phase passage 1. **b** Comparison of the phosphate solubilised by the selected parental communities (PB: $n = 6$ communities, PR: $n = 3$ communities) and selected propagated communities (OBP: $n = 14$ communities, ORP: $n = 7$ communities). Box plots: box spans 25th to 75th percentiles (IQR) with median as central line, whiskers cover values within 1.5 times IQR, and outliers as dots beyond whiskers. Letters in box plot: groups with different letters are statistically different based on pairwise comparison ($p$ values adjusted via Benjamini-Hochberg procedure). **c** Shifts in regression coefficients in communities of the parental and propagated offspring phase. *: Significances between performance. #: Significances between selection lines. Significance for the regression coefficient from GJAM obtained by the credible interval after the 10,000 MCMC iterations to estimate the values for each regression coefficient; whenever zero is within this 95% credible interval the regression coefficient is not considered as significant. **d, e, f** Community composition and abundance (CLR) of community members of the selected communities per cycle. **g** The residual correlation linked to solubilised phosphate. Relative closeness centrality is the highest for the ASV Citrobacter. Positive residual correlations shown as blue lines and negative residual correlations as red lines with the line thickness relating to the size of the residual correlations. PB parental best-performing communities, PR parental randomly selected communities. OBP propagated best-performing communities, ORP propagated randomly selected communities. Community members represented by ASVs and labelled with the most specific known taxonomic level. Source data are provided as Source Data file.

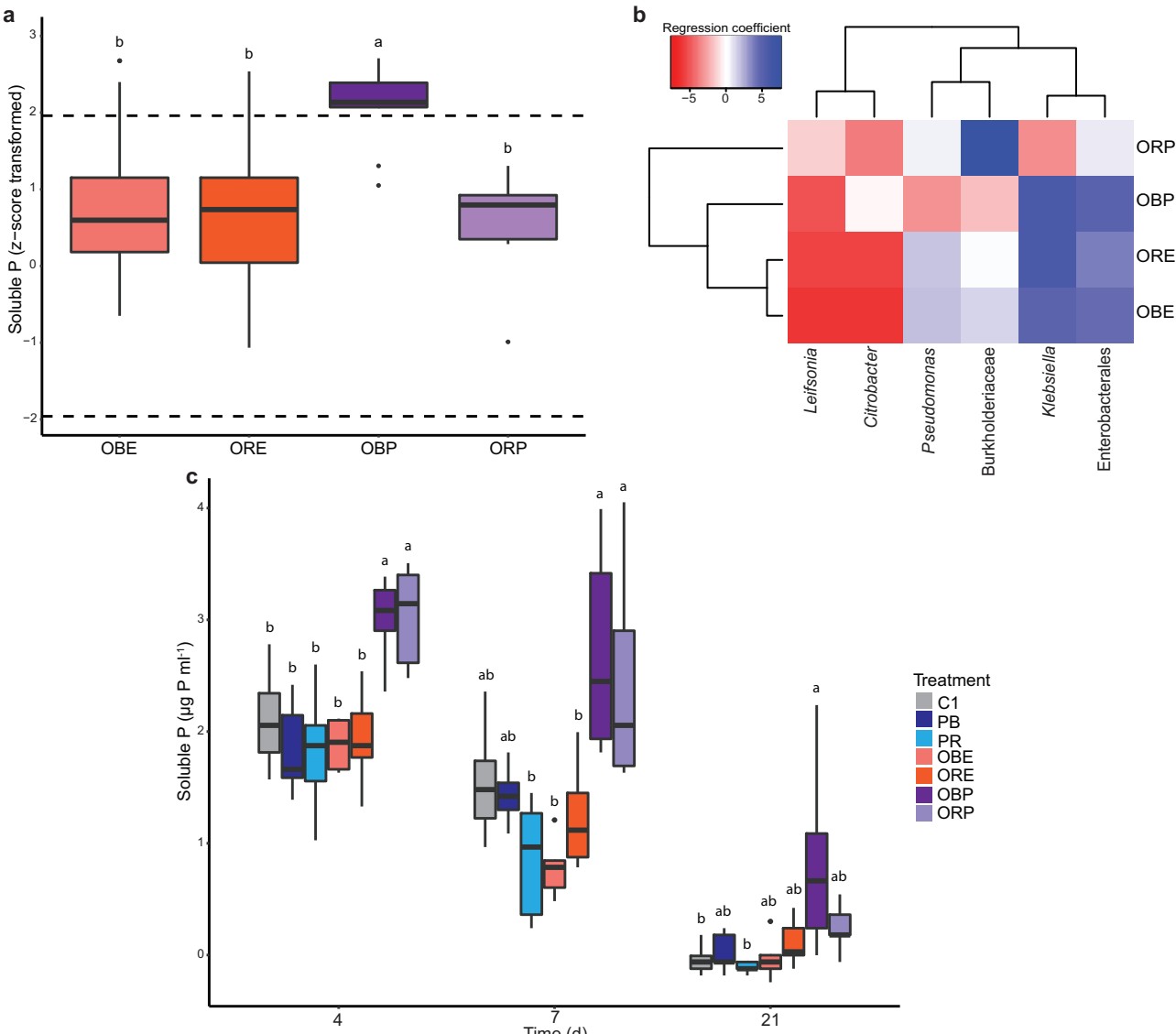

**Fig. 6 | Comparison of artificial selection methods and post-selection experiment. a** Phosphate solubilised by the offspring communities of the stable environmentally perturbed (OBE: $n = 144$ communities, ORE: $n = 72$ communities) and selected propagated communities (OBP: $n = 14$ communities, ORP: $n = 7$ communities). **b** Community composition between stable selected environmentally perturbed and propagated communities. Community members represented by ASVs and labelled with the most specific known taxonomic level. **c** Soluble phosphate available in the nutrient solution of the post-selection experiment. Treatments (for each treatment, $n = 6$ biologically independent replicates): Control: uninoculated nutrient solution containing insoluble iron phosphate. PB parental best-performing

community, PR parental randomly selected community. OBE environmentally perturbed best-performing community, ORE environmentally perturbed randomly selected community. OBP Propagated best-performing community, ORP propagated randomly selected community. Box plots: box spans 25th to 75th percentiles (IQR) with median as central line, whiskers cover values within 1.5 times IQR, and outliers as dots beyond whiskers. Letters in box plot per timepoint: groups with different letters are statistically different based on pairwise comparison (p values adjusted via Benjamini-Hochberg procedure). Source data are provided as Source Data file.

environmental perturbation phase communities, rarer community members had greater stability in the offspring propagation phase communities. Network analysis of residual correlations within the offspring propagation phase communities revealed two groups; however, both groups did not organise around the amount of solubilised phosphate (Fig. 5g). The first group, consisting of *Citrobacter*, Enterobacterales, and *Pseudomonas* showed mixed correlation with phosphate solubilisation, with only *Citrobacter* and Enterobacterales showing a positive association with soluble phosphate concentration. *Pseudomonas*, potentially due to its late emergence, clustered separately from other community members. The second group, including *Klebsiella*, *Burkholderiacea*, and *Leifsonia* displayed negative correlation with soluble phosphate concentration.

## Comparison of environmental perturbation and propagation

Scaled to the sterile controls, communities selected by propagation (z-score average ± SEM, 1.8 ± 0.3) solubilised phosphate on average 2.2-times higher than communities selected by environmental perturbation (z-score average ± SEM, 0.8 ± 0.1; Fig. 6a). The two-way ANOVA showed that phosphate solubilisation was significantly affected by the artificial selection method ($F(1,221) = 12.498$, $p < 0.001$; Supplementary Table 9), with propagated communities showing significantly higher soluble phosphate levels (two-way ANOVA + post hoc Tukey, $p < 0.001$; GLM + EMMs, $t(221) = 2.787$, $p = 0.006$; LMM + EMMs, $t(36.7) = 2.821$, $p = 0.008$). According to GLM and LMM analyses (but not ANOVA), actively selected communities from both methods solubilised significantly higher phosphate than randomly selected communities

**Table 1 | Comparison of the solubilised phosphate by the communities along the artificial selection phases**

| Community | Solubilised P [µg P ml⁻¹] in selection medium | Solubilised P (z-scores) in selection medium | Solubilised P [µg P ml⁻¹] in FeP NBRIP medium | Solubilised P [µg P ml⁻¹] in CaP NBRIP medium |
|---|---|---|---|---|
| PB | 378.05 ± 4.62 | 2.05 ± 0.40 | 58.40 ± 3.64 | 402.25 ± 4.62 |
| PR | 372.00 ± 3.49 | 1.66 ± 0.31 | 47.31 ± 3.64 | 284.27 ± 3.06 |
| OBE | 293.35 ± 10.91 | 3.52 ± 0.95 | 59.41 ± 9.62 | 360.90 ± 8.62 |
| ORE | 287.30 ± 1.75 | 2.99 ± 0.15 | 37.23 ± 1.01 | 242.93 ± 2.02 |
| OBP | 406.28 ± 14.65 | 3.91 ± 1.28 | 72.52 ± 13.86 | 346.79 ± 6.61 |
| ORP | 389.14 ± 5.61 | 2.78 ± 0.49 | 55.38 ± 4.40 | 284.27 ± 4.62 |

Data shown as mean ± standard error.
*PB* parental best-performing communities, *PR* parental randomly selected communities, *OBE* environmentally perturbed best-performing communities, *ORE* environmentally perturbed randomly selected communities, *OBP* Propagated best-performing community, *ORP* propagated randomly selected community.

(two-way ANOVA + Tukey, $p = 0.892$; GLM + EMMs, $t(221) = 3.969$, $p < 0.001$; LMM + EMMs, $t(36.7) = 4.340$, $p < 0.001$). Moreover, a significant interaction between the method and the type of selection line was observed (two-way ANOVA, $F(1,221) = 15.140$, $p < 0.001$), indicating that the impact of the method on phosphate solubilisation varied between active and random selection. Actively selected communities by propagation (OBP) outperformed both propagated, randomly selected communities (ORP) and communities selected by environmental perturbation (OBE and ORE) (two-way ANOVA + Tukey post hoc test & GLM + EMMs & LMM + EMMs, $p < 0.001$; Supplementary Table 10; Fig. 6a). Relative to the sterile control, OBP communities (z-score average ± SEM, 2.5 ± 0.2) exhibited phosphate solubilisation activity 5-times higher than ORP communities (z-score average ± SEM, 0.5 ± 0.3). Additionally, OBP communities displayed 3.1- and 2.8-times higher phosphate solubilisation activity compared to OBE (z-score average ± SEM, 0.8 ± 0.1) and ORE (z-score average ± SEM, 0.9 ± 0.1) communities, respectively.

When comparing the selected community compositions, the offspring environmental perturbed communities OBE and ORE showed the highest similarity to each other (Fig. 6b). The propagated communities OBP and ORP also clustered closely together but had lower abundance of ASVs belonging to *Pseudomonas* and Burkholderiaceae. The OBE community was the most diverse (Shannon index = 0.9), followed by the OBP communities (Shannon index = 0.65). In contrast, the ORP community was the least diverse (Shannon index = 0.01) and dominated by the ASV belonging to Burkholderiaceae, which sharply contrasted with other communities.

**Post-selection experiments**
Growing the selected communities in different phosphate-restricted media and conducting a plant hydroponic experiment allowed us to characterise their phosphate solubilisation activity post-selection (Fig. 1vi). The performance of selected communities in various phosphate-restricted media was superior to randomly selected communities. Both selection methods generated communities that solubilised more phosphate compared to the parental communities when provided with different iron phosphate media (Table 1). In the mixed media, the ORE community showed an 80.1% improvement, while the OBE showed community a 71.7% enhancement. Propagation led to a 90.7% increase in solubilisation activity for the OBP community, whereas the ORP community exhibited a 67.5% increase. In conditions with only insoluble iron phosphate, propagation yielded a 24.2% increase in solubilisation activity, while environmental perturbation resulted in a modest 1.7% increase. The ORP communities displayed a slight increase of 17.1%, whereas the solubilisation activity of the ORE

communities decreased by 21.3%. When exposed to only calcium phosphate ($Ca_3PO_4$), the parental communities demonstrated the highest solubilisation activity, surpassing all offspring communities.

In the plant hydroponics experiment, the inoculation of our communities did not have a significant effect on plant biomass (Supplementary Tables 11–13; Supplementary Fig. 1). However, the amount of solubilised phosphate differed between the treatments (Fig. 6c). The nutrient solution inoculated with the propagated communities exhibited significantly higher levels of soluble phosphate compared to the other inoculated treatments after four days (one-way ANOVA + post hoc Tukey, $p < 0.050$; GLM + EMMs, $p < 0.001$; Supplementary Tables 14 and 15). In comparison to the control treatment (C1), which provided only insoluble phosphate in the nutrient solution, treatments inoculated with communities of the parental phase showed a 14.3% reduction in soluble phosphate concentration. Treatments inoculated with environmentally perturbed communities exhibited a 9.5% reduction. In contrast, treatments inoculated with propagated communities showed, on average, a 42.9% increase in soluble phosphate compared to the C1 control. After seven days, the nutrient solutions inoculated with the propagated communities had significantly higher soluble phosphate levels than those inoculated with PR, OBE, and ORE communities (one-way ANOVA + Tukey & GLM + EMMs, $p < 0.050$). Furthermore, only treatments inoculated with communities selected by propagation increased soluble phosphate concentration, doubling the amount compared to the C1 control. Conversely, the parental phase communities and communities selected by environmental perturbation decreased the available phosphate by 6.7% and 26.7%, respectively, compared to the C1 control. These trends persisted at harvest, where only the best-performing propagated offspring communities (OBP) significantly elevated soluble phosphate levels compared to the C1 control treatment (one-way ANOVA + Tukey & GLM + EMMs: $t(30) = 3.432$, $p = 0.019$).

## Discussion
To increase the sustainability of agriculture, it is important to efficiently harness and modify microbial communities. In our study, we compared two methods of selecting microbial communities, namely environmental perturbation, and propagation, in their ability to enhance phosphate solubilisation activity. We found that dilution but not fractioning, led to a shift in community composition. Propagation was superior to environmental perturbation in generating microbial communities with enhanced phosphate solubilisation activity. Both artificial selection methods resulted in communities containing members positively and negatively associated with phosphate solubilisation. *Klebsiella* and Enterobacterales dominated most offspring communities. Moreover, the increased solubilisation activity observed in the propagated communities was transferable to a hydroponic plant system. Our findings suggest that propagation is better suited to consistently generate phosphate-solubilising microbial communities.

The solubilisation activity varied strongly during the stabilisation periods of the parental phase and the offspring environmental perturbation phase. The selected communities did not perform optimally in the first passage of the parental phase, confirming the inefficiency of prior stabilisation selection[32]. The reduction and fluctuation in phosphate solubilisation might be due to serial passaging, which is required for stabilisation. Through serial passaging, the most competitive community members are selected since there is no intercommunity selection that would promote the highest community function[28]. Consequently, this process facilitates the emergence of community members with reduced or lost activity (known as cheaters) who grow faster by avoiding the cost of contributing to community function. Furthermore, passaging presents a challenge due to reduced power in random sampling caused by the large size of the community[13]. By connecting the stability of the trait to community composition, we found that the stability of the taxa exhibited similarities with the

stability of the trait. This indicates that measuring the amount of solubilised phosphate as an indicator of community stability is feasible and offers the advantage of being quicker and less expensive than other proposed techniques, such as species richness[32,33]. Overall, our findings support previous research[32] in highlighting the necessity of stabilisation in the parental phase but also reveal a trade-off between selecting communities with peak performance and achieving stability.

Microbial communities selected by environmental perturbation had a more stable community composition compared to the parental phase. This increased stability may be attributed to the stronger influence of limited phosphate resources on community composition[34]. Our findings validate that environmental perturbation generates communities with altered activity[12,13]. However, these offspring communities did not show improvement in their phosphate solubilisation activity, which could be a result of an inadequate strength of the selection pressure. Another possibility is that the communities shifted their phosphate acquisition strategy, as bacteria in phosphate-limited soils prioritise efficient phosphate transport over solubilisation[35]. Another explanation could be that the serial passaging to establish communities favours intracommunity selection of cheaters[28]. Overall, we found that artificial selection through environmental selection pressure provides limited control over the performance of generated communities and is time-consuming due to repeated stabilisation periods.

Consistently, propagation generated significant phosphate-solubilising communities that outperformed randomly selected communities. Despite the repeated propagation, stability was maintained for four cycles, which aligns with recent findings suggesting that stability can be achieved and maintained during propagation as long as the environment remains stable[33]. However, stability decreased after the fifth cycle, likely due to the sudden emergence of an ASV from the *Pseudomonas* genus. This may indicate potential contamination from an external source since all selection lines exhibited a similar pattern. Furthermore, it is unlikely that the *Pseudomonas* ASV evolved into a cheater as the repeated intercommunity selection in the active selection lines should theoretically counteract such evolution[28]. Overall, artificial selection through propagation rapidly and consistently generated significant phosphate-solubilising communities.

The organisation and correlation with the amount of solubilised phosphate exhibited significant variation between the two selection methods. In the case of environmental perturbation, solubilised phosphate played a central role, whereas the propagated communities showed a greater distance from phosphate. This phenomenon could be explained that limited nutrients potentially play a more significant role in more restricted environments. Our offspring communities were primarily abundant in *Klebsiella* and Enterobacterales, which are well-known for harbouring the phosphate-solubilising trait[36,37]. Consistent with previous findings that indicate microbes can switch between phosphate solubilisation and consumption[38], none of our taxa were exclusively associated with either consuming or solubilising group.

Individual communities consistently outperformed the average community activity of the artificial selection phases in different phosphate growth media, highlighting the importance of intercommunity selection[28]. Similarly to previous selection experiments[24], our randomly selected communities also increased their activity. Interestingly, artificial selection did not enhance the ability of any offspring community to solubilise calcium phosphate, despite its higher water solubility and susceptibility to microbial solubilisation compared to iron phosphate[39]. We speculate that artificial selection promotes the specialisation of offspring communities towards specific phosphate sources, which could be attributed to the different pathways of phosphate solubilisation[6]. For instance, our selected communities may have adapted to siderophore secretion, which is not effective in solubilising calcium phosphate[39]. A similar trend of specialisation was achieved in a previous artificial selection experiment

involving microbial communities that conferred sodium salt tolerance[18]. Consequently, we recommend that in vitro selection for phosphate-solubilising communities should use the predominant phosphate source expected to be solubilised under field conditions. Further studies should explore the potential for adaptation and specialisation with other phosphate sources.

In conclusion, we compared the two artificial selection methods, namely environmental perturbation, and propagation, to determine their effectiveness in optimising the phosphate solubilisation activity of microbial communities. Our findings revealed that propagation consistently generated microbial communities with significant phosphate-solubilising capabilities, whereas environmental perturbation provided limited control over the performance of the generated communities. The study highlighted the importance of selection and stability before selection, aligning with previous studies[25,28]. Additionally, our post-selection experiments showed that the performance of the generated communities could be maintained when transferred to a hydroponic system, indicating a specialisation of the selected communities in their phosphate solubilisation activity. Considering our results, we have gained insights into more efficient methods for selecting plant growth-promoting communities, and we anticipate that in vitro artificial selection holds promise for successfully generating such communities.

## Methods

### Diversification of the source soil microbial by fractioning and diluting

To characterise the source soil microbial community, the microbial community from a natural grassland soil in Ede, The Netherlands (52.0445042, 5.6259190), was extracted by Nycodenz Density-Gradient separation with Tween20 and 80% Nycodenz[30]. To explore different soil matrix association strengths, we repeated the extraction process twice, creating two communities: easy-to-extract (EE), and hard-to-extract (HE). The soil pellet from the second extraction was resuspended in phosphate-buffered saline and called the not-extractable (NE) community (Fig. 1i).

We further diversified the communities using dilutions[29] (Fig. 1ii). The EE and HE communities were combined, creating the undiluted community (D0), and diluted by 1:2 (v:v, D1) and by 1:4 (v:v, D2). For loss of abundance was controlled by centrifuging (20 min, 10,000 g) and removing the supernatant according to each dilution (i.e., D0 = 0%, D1 = 50%, D2 = 75%).

### Selection of phosphate-solubilising microbial communities

To enrich for phosphate solubilisation, communities were cultivated in an adapted National Botanical Research Institute's Phosphate (NBRIP) growth medium[40], supplying $FePO_4 \cdot 2H_2O$ as the insoluble phosphate source. The medium was adapted for the different selection methods; in the parental phase and offspring propagated phase, we supplemented the medium with 1 g $KH_2PO_4$. In the offspring environmentally perturbed phase, we reduced the amount of $KH_2PO_4$ to 0.75 g to create an environmental selection pressure. All communities were incubated in the dark (72 h, 28 °C, 180 rpm) and passaged by transferring 5% of the preceding community in fresh medium. The soluble phosphate was measured in all communities after the incubation using the vanado-molybdo-phosphoric yellow colour method[41]. Briefly, after centrifuging the communities (20 min, 10,000 g), 4 μl of culture supernatant were mixed with 186 μl $dH_2O$ and 10 μl vanadate-molybdate reagent in a microplate. After 10 min incubation, the concentration of soluble phosphate was measured spectrophotometrically at 430 nm.

To start the parental phase, the fractioned and diluted communities were incubated in triplicates (Fig. 1iii). The offspring environmentally perturbed phase was initiated with the two best-performing stable parental communities from each fractioned and

diluted community resulting in six active selection lines, and three randomly selected communities resulting in three random selection lines with eight replicates per selection line (Fig. 1iv). The parental phase and the offspring environmental perturbation communities were stabilised by serial passaging without selection[13], using the solubilised phosphate as a proxy for community stability. Communities reached stability when the amount of soluble phosphate between passages was stable, which was determined with a change point analysis using the Pruned Exact Linear Time (PELT) approach[42].

The offspring propagation phase was started using the two best-performing parental communities resulting in two active selection lines (B1 and B2), and one randomly chosen community as control, with 12 replicates per selection line (Fig. 1v). After each incubation, the community with the most soluble phosphate was selected to start the next cycle in the active selection lines. In the random selection line, the communities to start the next cycle were randomly chosen. This procedure was repeated for seven cycles.

## Post-selection experiments

For the first post-selection experiment, the solubilisation activity of the selected communities was tested in different phosphate-restricted media (Fig. 1vi). For this, 100 μl of the selected communities were grown in 10 ml of the NBRIP media where the communities were selected, and NBRIP media without soluble phosphate, containing either $Ca_3PO_4$ or $FePO_4 \cdot 2H_2O$ as the only phosphate source. The solubilised phosphate was measured for all communities.

For the second post-selection experiment chrysanthemum (*Chrysanthemum indicum* L., cultivar Chic Cream) cuttings were grown in a hydroponic system. First, the chrysanthemum cuttings were allowed to grow roots for 6 days in sterile 0.5 Hoagland solution[43]. Afterwards, the roots were rinsed with sterile $dH_2O$. Six replicates for each treatment with three cuttings per replicate were transferred into 150 ml sterile 0.5 Hoagland solution substituted with 0.5 mM insoluble $FePO_4 \cdot 2H_2O$. The volume was maintained at 150 ml throughout the experiment. Depending on the treatment, the nutrient solution was inoculated with 1 ml of either the parental best (PB), parental random (PR), environmentally perturbed offspring best (OBE), environmentally perturbed offspring random (ORE), propagated offspring best (OBP) or propagated offspring random (ORP) communities. To prepare the inocula, 100 μl of the respective community was grown in 10 ml of the media where the communities were previously selected. Furthermore, two sterile, uninoculated control treatments were included, providing 0.5 mM insoluble $FePO_4 \cdot 2H_2O$ in control treatment 1 (C1), and 0.5 mM soluble $KH_2PO_4$ in control treatment 2 (C2). The order of the treatments was randomised according to a Randomised Complete Block Design. The plants were grown in a growth cabinet (28 °C/28 °C day/night, 16 h d$^{-1}$ light) for 21 days, after which the plants were harvested and separated into shoots and roots for fresh and dry weight determination. The amount of soluble phosphate in the nutrient solution was quantified on days 4, 7 and 21 of the growth periods.

## DNA extraction and 16S rRNA partial gene sequencing and bioinformatic analyses

The 16S rRNA partial gene amplicon sequencing was performed on samples from all fractions and dilutions of the source community, and the selected best and random performing communities of all passages and cycles of the artificial selection phases. The DNA was extracted using the DNeasy PowerSoil Kit (QIAGEN, Germany). The samples were sent to Genome Québec (Québec, Canada) for sequencing on the Illumina MiSeq system, generating 250 base pairs (bp) long paired-end reads targeting the V4 regions gene with the primers 515 F (forward primers – 5'-CAAGTGCCAGCMGCCGCGGTAA-3') and 806 R (reverse primers – 5'-CATGGACTACHVGGGTWTCTAAT-3'). The reads of the 16S rRNA partial gene amplicon sequencing were processed using the DADA2 pipeline[44]. The taxonomy was assigned to the amplicon sequencing variants (ASVs) with the SILVA database (v138)[45]. The DADA2-processed sequencing data was filtered to remove unassigned ASVs.

## Statistical analysis

Statistical analysis was performed using R v4.2.0[46]. Community stability was approximated by the stability of solubilised phosphate across the passages. A community was considered stable if only minimal variation occurred over at least three passages, which were referred to as islands of stability. To compare communities from different selection backgrounds, phases, passages, and cycles, z-score transformed data comparing to the respective sterile controls was used. Z-score values between −1.96 and 1.96 fell into the 95% confidence interval from the average. Communities with z-scores >1.96 were considered significant phosphate solubilisers. To meet the assumption of normal distribution for statistical analysis, the data were transformed, and outlying values identified via Tukey's fences were excluded when required. Levene's test[47] was used to test for heterogeneity of variance in the phosphate solubilisation activity of the fraction and dilution communities in the first parental passage. The heterogeneity of the data was accounted for the multiple comparison tests, when needed, via an adapted variance-covariance matrix for heteroscedasticity-consistent (HC) covariances in linear models[48]. Analysis of variance (ANOVA) followed by Tukey post hoc test (ExpDes (v1.2.2), multcomp (v1.4.25) packages[49,50]) with a significance level of 5% was used for statistical comparisons. Additionally, generalised linear models (GLM) and linear mixed-effects models (LMM, lme4 package v1.1.34[51]) were used for further statistical comparison and followed with estimated marginal means (EMMs, emmeans package 1.8.9[52]) as post hoc pairwise comparison with a significance level of 5%. In the LMMs, the variability across phases or cycles was accounted for each replicate by specifying them as a fixed random factor. In the plant hydroponics experiment, we accounted for the block as a fixed factor in both ANOVA and GLM. All *p* values were corrected for multiple comparison using the Benjamini-Hochberg procedure to address the false discovery rate of multiple comparison tests[53]. The results of the various statistical models and post hoc analyses are compared in the Supplementary Tables 1–15.

The effect of artificial selection on the communities was analysed using generalised joint attribute modelling (GJAM, gjam package v2.6.2)[54]. This allowed us to combine different types of variables, in our case the compositional constrained community data and the continuous phosphate solubilisation, to evaluate the effects of our complex experimental design (different dilutions or fractions x phase x artificial selection method). The timepoints (passages and cycles) were specified as random factor. For this, ASVs present in fewer than 5 samples were aggregated into a single class (others). In the GJAM model, the passages were set as a random factor to account for the lack of independence between the samples. The regression coefficients provided by the model were used to show how the different variables responded to the explanatory variables. The regression coefficients were estimated from the Gibbs sampling that was set to 10,000, discarding the first 4000 iterations as they do not provide meaningful estimation. Gibbs estimates were also used to obtain the 95% confidence intervals, and regression coefficients with confidence intervals that excluded zero were considered significant. Significant shifts in the microbial communities were visualised using heatmaps that were scaled by columns (gplots package v3.1.3)[55]. Oscillation of the ASVs per passage was analysed using sequencing data that was centre log-ratio (CLR) transformed with Bayesian-multiplicative replacement of count zeros (CZM)[56]. Shannon diversity was calculated using DivNet[57], a method that adjusts the estimates of Shannon diversity to account for the network structure of the microbial community and the compositional nature of the data. The associations between ASVs and soluble

phosphate were extracted from the GJAM residual correlation matrices. These residual correlations correspond to the association between two dependent variables that are not explained by the experimental design, which is usually interpreted as potential biotic interaction in community ecology[58,59]. The residual correlation matrices were further evaluated for its network organisation using centrality measurements from network analysis (qgraph package v1.9.5)[60].

## Reporting summary
Further information on research design is available in the Nature Portfolio Reporting Summary linked to this article.

## Data availability
The raw sequencing data generated in this study have been deposited in the European Nucleotide Archive (ENA) database under accession code PRJEB64328. The taxonomy was assigned to the amplicon sequencing variants (ASVs) with the SILVA database (v138) (https://www.arb-silva.de/no_cache/download/archive/release_138/). The source data generated in this study are provided as a Source Data file with this paper. Source data are provided with this paper.

## Code availability
The R code for statistical analysis and GJAM modelling is provided in the Supplementary information file titled Supplementary code.

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

## Acknowledgements

This work was supported by the Top Consortium for Knowledge and Innovation, TKI Agri & Food (TU 17008 TKI, E.E.K.). We thank Royal Van Zanten BV for providing the chrysanthemum cuttings and Koppert Biological Systems BV for project support.

## Author contributions

L.F., M.F.A.L. and E.E.K. planned and designed the research; L.F. executed the experiments; L.F. and M.F.A.L. analysed the data; L.F. draughted the manuscript. E.E.K. revised the manuscript. All authors have read and approved the submitted version.

## Competing interests

The authors declare no competing interests.
