## [Peer Review File · Nature Communications]

Reviewers' Comments:

Reviewer #1:

Remarks to the Author:

The idea behind this manuscript is very novel and interesting, and I like how experiments were planned. Unfortunately, I do not think that this version is ready for publication for three major reasons: lack of statistical evidence, statements not supported by results, and weaknesses in manuscript organization.

Lack of statistical evidence. I was very surprised to notice that statistical evidence is basically lacking throughout the results section. Yes, authors report some p-values here and there, but full statistical details are completely missing, together with details on the statistical models in more complex analyses. At the current state, it is impossible to assess (i) if the right tests were used, (ii) whether claims are supported by evidence (see my next point).

Statements not supported by results. Following my point above, I found several statements to be contradictory with the data shown. For example, claims at L93, L102, L103, and L104 contradict Figs 1 and 2, where no differences between groups were shown.

Manuscript organization can be improved. I think that the current organization of the manuscript does not fit the format required by a IRDaM journal. I think authors need to provide several information upfront, even with few details, allowing authors to grasp the results without constantly looking at the methods. For example, EE, HE, and NE need to be defined upfront. Similarly, authors need to explain why communities were diluted, and in general the rationale behind each experimental step.

In addition, I find that not including a negative control (no inoculation, or inactivated inoculum) in the last experiment (L316) is a major weakness.

In addition to my major concerns above, I also think that authors used wrong terms in several occasion:

- generation. This is a tricky one, as we use this term usually referring to generations as outcome of reproduction in single species. Here you have communities, made of species with very different generational times, so I think that generation is not the correct term to use. I'd instead opt for cycle.
- elite communities. The term elite is often used in genetics, and still I think it is inappropriate when dealing with communities.

I would also suggest to move the last figure as Fig. 1, and to make available data and code (all of it), so reviewers (and then readers) can have a deeper look at your findings.

Reviewer #2:

Remarks to the Author:

Dear Editor,

I appreciate the opportunity to evaluate the manuscript by Faller et al., which employed two artificial selection approaches to generate stable and enhanced phosphate-solubilizing microbial communities. This study holds significance in microbiome research, specifically in its application of a top-down approach encompassing the entire microbial community to select for targeted functional groups, such as phosphate-solubilizers. The manuscript is well-structured, yet it requires clarification, improved presentation, and precise articulation. Below, I have outlined both major and minor points for your consideration:

Major Comments:

1. Clarification is needed regarding the classification criteria for "easy-to-extract (EE)" and "hard-to-extract (HE)" categories.

2. The rationale for employing Chrysanthemum cuttings in the plant hydroponic system and its relevance to phosphate-solubilizing microbial communities should be clearly elucidated.
3. In the statistical comparisons on page 4, lines 88-97, specify whether the comparisons are among EE, HE, and NE (D0, D1, D2), or if they are between each pair separately
4. Line 102-103. "The NE communities had the highest average soluble phosphate concentration". Based on figure 2a, the differences among EE, HE and NE are not significant. Please make sure the expression here is accurate. This goes the same for line 104, line 106. Please double check.
5. Ensure proper referencing of figures and tables when presenting data, addressing instances like lines 102, 105, 110, 111, 129-130, 132-133, 139-141, 145-146, 161-162, and 181-183.
6. Standardize abbreviations for consistency, particularly in cases such as OBE (offspring best performing community) and OBP (unspecified abbreviation). First mentions of abbreviations should include their full names, e.g., SL (mention in line 128); OBP (line 152)
7. Clarify the identity of "rare community members" mentioned in line 117-120, especially since Pseudomonas is noted in line 120 and is not typically considered rare.
8. Line 177-178, the ORP is not significantly higher phosphate solubilization compared to the communities generated through environmental perturbation.
9. Include plant biomass data, if available, in supplementary format (line 197-198).
10. Line 218, what types of cheaters the authors refer to? Why?
11. Line 244, why this indicates contamination? Please specify.

Minor Comments:

1. Line 251, please remove the extra "known"
2. Correct the spelling error in Figure 6: "perpetuation" to "perturbation."

REVIEWER COMMENTS

Reviewer #1 (Remarks to the Author):

The idea behind this manuscript is very novel and interesting, and I like how experiments were planned. Unfortunately, I do not think that this version is ready for publication for three major reasons: lack of statistical evidence, statements not supported by results, and weaknesses in manuscript organization.

Lack of statistical evidence. I was very surprised to notice that statistical evidence is basically lacking throughout the results section. Yes, authors report some p-values here and there, but full statistical details are completely missing, together with details on the statistical models in more complex analyses. At the current state, it is impossible to assess (i) if the right tests were used, (ii) whether claims are supported by evidence (see my next point).

R: We apologize for the missing details. We improved the methods by including a section with detailed statistical analysis (L384-419) and by providing which test was used together with all provided p-values.

Statements not supported by results. Following my point above, I found several statements to be contradictory with the data shown. For example, claims at L93, L102, L103, and L104 contradict Figs 1 and 2, where no differences between groups were shown.

R: Thank you for the opportunity to clarify these points. As mentioned by the reviewer, the analysis performed on the undiluted samples (former lines 91-94) showed no significant difference between the different fractions or dilutions based on the Tukey test. However, as we have now better articulated, we present significant effects obtained from the GJAM model output (new Fig. 2d and L96-99). We have clarified that the differences were not detected by the Tukey test.

Regarding the results of the parental phase analysis (previously described in lines 101-105), we referred to z-score values from different replicates to report the percentage of communities exhibiting significantly higher P-solubilization compared to the control treatment (the non-inoculated control). Despite not finding a general trend for the community fractions or dilutions, we identified groups of microbes exhibiting P-solubilization surpassing the control treatment. These groups could potentially be selected to artificial selection trials. We have provided a more detailed explanation of these findings in the Results section whenever the z-score was mentioned and have also enhanced the explanation of the methods in the new Statistical Analysis section.

Manuscript organization can be improved. I think that the current organization of the manuscript does not fit the format required by a IRDaM journal. I think authors need to provide several information upfront, even with few details, allowing authors to grasp the results without constantly looking at the methods. For example, EE, HE, and NE need to be defined upfront. Similarly, authors need to explain why communities were diluted, and in general the rationale behind each experimental step.

R: To improve readability, we included more information and added the rationale about the selection methods, fractions, and dilutions in the introduction (L63-69: "propagation involves the repeated seeding of new offspring communities from random subsets of each selected community based on the desired community function, favouring high-performing communities through intercommunity

selection. Moreover, this study seeks to explore the ecology of a cultivatable phosphate-solubilising community by combining dilutions and soil community fractioning. Fractioning by repeated Nycodenz extraction has been suggested to result in communities with varying degrees of association with the soil matrix, while serial dilution provides insights into how community complexity influences phosphate solubilisation.”).

We also included a short introduction sentence in each result section to explain what was done and why (L82-85, L92-94, L133-135, L163-164, L207-209). Furthermore, as suggested later by the reviewer, we changed the order of the figures after moving the workflow picture as Fig. 1 and referred to it in the topic sentences of the respective result sections.

In addition, I find that not including a negative control (no inoculation, or inactivated inoculum) in the last experiment (L316) is a major weakness.

R: In fact, the controls are included. While we showed the control data (sterile nutrient solution containing insoluble iron phosphate) in Fig. 5c, we now better detailed it in the Methods section. We addressed this comment by describing the 2 control treatments (sterile Hoagland solution with insoluble P, sterile Hoagland solution with soluble P) in the Materials and Methods section (L366-368: “Furthermore, two sterile, uninoculated control treatments were included, providing 0.5 mM insoluble $\text{FePO}_4 \cdot 2\text{H}_2\text{O}$ in control treatment 1 (C1), and 0.5 mM soluble KH_2PO_4 in control treatment 2 (C2).”).

In addition to my major concerns above, I also think that authors used wrong terms in several occasion:

- generation. This is a tricky one, as we use this term usually referring to generations as outcome of reproduction in single species. Here you have communities, made of species with very different generational times, so I think that generation is not the correct term to use. I'd instead opt for cycle.

R: After careful consideration we substituted the word 'generation' throughout the manuscript: for the propagated communities, where we selected after each incubation, we agree in using the suggested 'cycle' instead of 'generation'.

However, we did not see how the proposed 'cycle' would fit in the context of the 'parental and environmental perturbation' experimental phase. Therefore, we decided to call these experimental blocks separated by selection 'phase'. Moreover, within the former 'generations' we defined the serial passages without manual selection as 'cycles'. In this context, we replaced the 'cycle' with 'passage'.

We adapted this new terminology throughout the manuscript and adjusted the figures accordingly.

- elite communities. The term elite is often used in genetics, and still I think it is inappropriate when dealing with communities.

R: Thanks. Instead of elite, we now use active selection line/actively selected community. We think that this new term is well suited since these communities are actively chosen, which highlights the contrast to the randomly selected control (random selection lines and communities). We adapted this new terminology throughout the manuscript and adjusted the figures accordingly.

I would also suggest to move the last figure as Fig. 1, and to make available data and code (all of it), so reviewers (and then readers) can have a deeper look at your findings.

R: The last figure is moved to Figure 1 as suggested. All the R scripts and codes are submitted together with the revised manuscript.

Reviewer #2 (Remarks to the Author):

Dear Editor,

I appreciate the opportunity to evaluate the manuscript by Faller et al., which employed two artificial selection approaches to generate stable and enhanced phosphate-solubilizing microbial communities. This study holds significance in microbiome research, specifically in its application of a top-down approach encompassing the entire microbial community to select for targeted functional groups, such as phosphate-solubilizers. The manuscript is well-structured, yet it requires clarification, improved presentation, and precise articulation. Below, I have outlined both major and minor points for your consideration:

Major Comments:

1. Clarification is needed regarding the classification criteria for “easy-to-extract (EE)” and “hard-to-extract (HE)” categories.

R: As both reviewers suggested, we clarified the rationale behind fractioning in the Introduction (L65-69: “Moreover, this study seeks to explore the ecology of a cultivatable phosphate-solubilising community by combining dilutions and soil community fractioning. Fractioning by repeated Nycodenz extraction has been suggested to result in communities with varying degrees of association with the soil matrix, while serial dilution provides insights into how community complexity influences phosphate solubilization.”) We further introduced the fractioned communities in the topic sentence of the respective paragraph in the result section (L82-85: “To explore different soil matrix association strengths, we performed repeated Nycodenz extraction to create three fractions, easy-to-extract (EE), hard-to-extract (HE), and not-extractable (NE), that allowed us to explore the ecologically relevant fraction of phosphate solubilisers within a soil microbial community (Fig. 1i).”). And finally, the classification criteria, how we obtained and labelled the communities, is described in materials and methods section L317 to L320.

2. The rationale for employing Chrysanthemum cuttings in the plant hydroponic system and its relevance to phosphate-solubilizing microbial communities should be clearly elucidated.

R: We used chrysanthemum since it is a model plant in our group. Furthermore, chrysanthemum has high economic value, which we introduced by mentioning that we used “a commercial chrysanthemum (*Chrysanthemum indicum* L.) cultivar” (L75-76). Chrysanthemum is also an agricultural system that heavily relies on inorganic nutrient input. With that in mind our study also envisions a potential use of the microbial communities to promote a more efficient use of soil phosphate (L75-76).

3. In the statistical comparisons on page 4, lines 88-97, specify whether the comparisons are among EE, HE, and NE (D0, D1, D2), or if they are between each pair separately

R: Thanks for the suggestion. To increase clarity, we added that we used a one-way ANOVA followed by a Tukey post hoc test (L86; L95-96). We also better detailed the results from the GJAM model in line with the suggestions of the Reviewer #1 (L96-99). By including this, it should be clearer that we compared each pair separately, since we used the Tukey post hoc test.

4. Line 102-103. "The NE communities had the highest average soluble phosphate concentration". Based on figure 2a, the differences among EE, HE and NE are not significant. Please make sure the expression here is accurate. This goes the same for line 104, line 106. Please double check.

R: We referred to z-score values from different replicates to report the percentage of communities exhibiting significantly higher P-solubilization compared to the control treatment (the non-inoculated control), now specified in L108, and L111. Despite not finding a general trend for the community fractions or dilutions, we identified groups of microbes exhibiting P-solubilization surpassing the control treatment. These groups could potentially be selected to artificial selection trials. We have provided a more detailed explanation of these findings in the Results section whenever the z-score was mentioned and have also enhanced the explanation of the methods in the new Statistical Analysis section.

5. Ensure proper referencing of figures and tables when presenting data, addressing instances like lines 102, 105, 110, 111, 129-130, 132-133, 139-141, 145-146, 161-162, and 181-183.

R: We checked for proper referencing of figures and tables throughout the manuscript and adjusted accordingly. For several lines mentioned in this comment, the data was only presented within the result section and not in figures or tables (e.g., Shannon indices, or exact amounts of soluble phosphate at the point of selection), or it was already referenced in the previous sentence.

6. Standardize abbreviations for consistency, particularly in cases such as OBE (offspring best performing community) and OBP (unspecified abbreviation). First mentions of abbreviations should include their full names, e.g., SL (mention in line 128); OBP (line 152)

R: We checked all abbreviations and made sure to include their full names in all cases when we first mentioned them. To make the manuscript clearer, we removed some of the abbreviations and kept the full terms. For clarity and easy to the readers to follow our experiments, the new Figure 1 illustrates all the abbreviations used in the manuscript.

7. Clarify the identity of "rare community members" mentioned in line 117-120, especially since *Pseudomonas* is noted in line 120 and is not typically considered rare.

R: To clarify, with rare community members we referred to ASVs with a CLR-transformed abundance smaller than zero. In the previous L120, we mentioned the great oscillation of *Pseudomonas*, *Citrobacter*, *Leifsonia* and *Enterobacteriaceae*, because they go from very low abundance to high abundance, but we do not classify them as rare. To prevent a mix up of these two separate statements,

and since the oscillation between parental and environment perturbed offspring communities are compared later (L154-156), we decided to take this sentence (previously 117) out.

8. Line 177-178, the ORP is not significantly higher phosphate solubilization compared to the communities generated through environmental perturbation.

R: Yes, indeed ORP (the randomly selected propagated communities) are not higher. However, in the mentioned lines, we reported that the propagated ELITE (now re-named upon reviewer request to “active”) selection line communities are higher (L164-167). To improve clarity, we provided the abbreviation of the selection lines (OBP1 & OBP2) in the text.

9. Include plant biomass data, if available, in supplementary format (line 197-198).

R: Thanks for the suggestion. We have included plant fresh and dry biomass data as Figure S1 in the supplementary material.

10. Line 218, what types of cheaters the authors refer to? Why?

R: We included a definition of cheaters and explained why they can emerge and how. See L243-L248: “The reduction and fluctuation in phosphate solubilisation might be due to serial passaging, which is required for stabilisation. Through serial passaging, the most competitive community members are selected since there is no intercommunity selection that would promote the highest community function²⁸. Consequently, this process facilitates the emergence of community members with reduced or lost activity (known as ‘cheaters’) who grow faster by avoiding the cost of contributing to community function.”

11. Line 244, why this indicates contamination? Please specify.

R: We discussed this more extensively why we think it could indicate a contamination in L272-275: “This may indicate potential contamination from an external source since all selection lines exhibited a similar pattern. Furthermore, it is unlikely that the *Pseudomonas* ASV evolved into a cheater as the repeated intercommunity selection in the active SLs should theoretically counteract such evolution”

Minor Comments:

1. Line 251, please remove the extra “known”

R: Removed.

2. Correct the spelling error in Figure 6: "perpetuation" to "perturbation."

R: Corrected (now Figure 1).

Reviewers' Comments:

Reviewer #1:

Remarks to the Author:

I'd like to thank the author for working on the manuscript following reviewer's suggestions.

While the manuscript certainly improved, I still think that the evidence authors provide for the results is still weak. Authors still not report the full results from the data analysis, but only whether the p-value was higher or lower than 0.05. This does not tell the readers the whole story, and it might make people wonder if the correct approach was used.

In addition, when more than one factor is tested, authors still use ANOVA. Considering that biological data rarely follows the assumptions necessary to correctly perform an ANOVA, and that often the authors deal with a complex design, I'd suggest the authors to use a different type of linear model, such a `lm/glm` or `lmer/glmer`. In this way authors can also account for random factors. Also, while Tukey's test is great, I think their analysis can benefit from using pairwise contrasts (e.g., `emmeans`), correcting the p-values for multiple comparisons.

Finally, Tab. 1 still lacks either SE or SD for mean values.

Reviewer #2:

Remarks to the Author:

The authors have restructured the manuscript, incorporated adequate statistical analysis methods, and addressed my concerns with greater clarity. I am content with the revisions made by the authors.

Responses to reviewer #1

I'd like to thank the author for working on the manuscript following reviewer's suggestions.

While the manuscript certainly improved, I still think that the evidence authors provide for the results is still weak. Authors still **not report the full results from the data analysis**, but only whether the p-value was higher or lower than 0.05. This does not tell the readers the whole story, and it might make people wonder if the correct approach was used.

R: We expanded and detailed the descriptions of the results to provide a more in-depth narrative of our findings, including the new results based on different types of models suggested by the reviewer. The expanded descriptions are in lines 85-90; 137-154; 185-195; 218-237; 263-279 in purple color.

We also provide the complete reports of the new statistical analyses, as suggested by the reviewer, in Supplementary Tables 1 to 15.

In addition, when more than **one factor is tested, authors still use ANOVA**. Considering that biological data rarely follows the assumptions necessary to correctly perform an ANOVA, and that often the authors deal with a complex design, I'd suggest the authors to **use a different type of linear model, such a lm/glm or lmer/glmer**. In this way authors can also account for random factors. **Also, while Tukey's test is great, I think their analysis can benefit from using pairwise contrasts (e.g., emmeans), correcting the p-values for multiple comparisons.**

R: We have reanalysed all our data using ANOVA + Tukey and the suggested GLM + EMMs. We also performed analyses with the suggested LMM + EMMs, considering fixed and random factors as suggested by the reviewer, particularly when comparing multiple passages/cycles and selection lines. When employing ANOVA, we ensured that the assumptions of normal distribution and heterogeneity of variances were met, as described in the Material and Methods section (Lines 443-445).

Following the reviewer's suggestion, we computed pairwise comparisons of Estimated Marginal Means (EMMs). All *p* values were corrected for multiple comparisons using the Benjamini-Hochberg procedure. Subsequently, we updated the Material and Methods section on statistical analysis accordingly (Line 450-458).

The results obtained from GLM + EMMs and LMM + EMMs show high similarity to the results from ANOVA + Tukey. However, there are instances where GLM/LMM + EMMs yielded *p* values <0.05, while the corresponding ANOVA + Tukey *p* values were not significant. We describe these results in lines 193-195, and provide a direct comparison of the results from the different models and post hoc comparisons in the Supplementary Material (Supplementary Tables 1-15).

Finally, Tab. 1 still **lacks either SE or SD for mean values.**

R: We provided the SE values in the Table 1.

Response to reviewer #2

The authors have restructured the manuscript, incorporated adequate statistical analysis methods, and addressed my concerns with greater clarity. I am content with the revisions made by the authors.

R: Thank you.

Reviewers' Comments:

Reviewer #1:

Remarks to the Author:

Thanks to the authors for providing a revised version of their manuscript. All my concerns have been addressed.

REVIEWERS' COMMENTS

Reviewer #1 (Remarks to the Author):

Thanks to the authors for providing a revised version of their manuscript. All my concerns have been addressed.

Response: Thank you for the positive response.